# Hypertension in the United States Fire Service

**DOI:** 10.3390/ijerph18105432

**Published:** 2021-05-19

**Authors:** Saeed U. Khaja, Kevin C. Mathias, Emilie D. Bode, Donald F. Stewart, Kepra Jack, Steven M. Moffatt, Denise L. Smith

**Affiliations:** 1Advocate Lutheran General Hospital, Park Ridge, IL 60068, USA; skhaja@hpil.org; 2Medical Advisor, Hanover Park Fire Department, Hanover Park, IL 60133, USA; 3Clinical Cardiac Electrophysiology, University of Illinois at Chicago, Chicago, IL 60612, USA; 4Health and Human Physiological Sciences, Skidmore College, Saratoga Springs, NY 12866, USA; kmathias@skidmore.edu (K.C.M.); ebode@skidmore.edu (E.D.B.); 5Public Safety Occupational Health Center, Fairfax, VA 22030, USA; dstewart97@yahoo.com; 6HeartFit For Duty, LLC, Mesa, AZ 85206, USA; kepra@heartfitforduty.org; 7Public Safety Health Systems, Ascension St. Vincent, Indianapolis, IN 46260, USA; steven.moffatt@ascension.org

**Keywords:** blood pressure, firefighting, firefighters, cardiovascular disease

## Abstract

Hypertension is a major risk factor for atherosclerotic cardiovascular disease and cardiac remodeling and is associated with an increased risk of sudden cardiac events, the leading cause of duty-related death in the fire service. We assessed systemic blood pressures and prevalence of hypertension among US firefighters by decade of life. Medical records of career firefighters (5063 males and 274 females) from four geographically diverse occupational health clinics were assessed. Hypertension was defined as systolic blood pressure ≥130 mmHg or diastolic blood pressure ≥80 mmHg, or taking antihypertensive medication. Results from the firefighter sample were compared to the US general population (2015–2016 and 2017–2018 National Health and Nutrition Examination Surveys). Among the total sample, 69% of firefighters met the criteria for hypertension and 17% were taking antihypertensive medications. Percentages of hypertensive male and female firefighters were 45% and 11% among 20–29 years old, respectively, and increased to 78% and 79% among 50–59 years old, respectively. Compared to the general population, male firefighters had a higher prevalence of hypertension (*p* < 0.05) across all age groups (11–16% higher). In order to improve firefighter health and protect against sudden incapacitation in this public safety occupational group, increased efforts are necessary to screen for and manage high blood pressure.

## 1. Introduction

Sudden cardiac events have consistently accounted for approximately 50% of duty-related deaths among firefighters over the last decade [1,2]. Furthermore, firefighters are more likely to suffer a sudden cardiac event following fire suppression activities than during routine activities at the fire station, clearly establishing that the cardiac events can be triggered by the work itself [3,4]. A retrospective study examining all available autopsies from firefighter fatalities over the past 20 years found that 80% of cardiac-related fatalities had both coronary heart disease and a structurally enlarged heart (cardiomegaly and/or left ventricular hypertrophy) [5].

Hypertension is among the most powerful modifiable risk factors for cardiovascular disease (CVD), mortality, and disability in all populations [6,7]. Further, hypertension was associated with a 12-fold increased risk for acute coronary event-related deaths among firefighters [8]. Longstanding hypertension leads to progressive cardiac remodeling [6,9,10], and both left ventricular hypertrophy (LVH) and cardiomegaly are associated with increased risk of sudden cardiac death in the fire service [4,5]. Recent evidence suggests that, within the same blood pressure category, firefighters may have greater risk of a sudden cardiac event than the general population [11]. Firefighters are also exposed to multiple occupational stressors that may increase their likelihood of developing hypertension, including high concentration of smoke and particulate matter [12,13], psychological stress [14], noise [15], and disrupted sleep [16].

To date, studies on hypertension and other CVD risk factors in firefighters have generally reported on small sample sizes, limited geographical regions, a select group of firefighters, and mainly males [17,18,19,20,21,22,23]. Currently, only one study has investigated blood pressure and hypertension across different age groups of firefighters [24], and studies that have compared hypertension in the fire service to the general population have found mixed results and were based on small samples [17,19,20,24]. Therefore, the purpose of this study was to examine blood pressure values and antihypertensive medication usage by decade of life in firefighters and among the general population.

## 2. Methods

### 2.1. Study Sample

Medical records from occupational health clinics that perform periodic medical evaluations for firefighters based in southern Arizona “Southwest cohort”, northern Virginia “Mid-Atlantic cohort”, central Florida “Southeast cohort”, and the capital region of Indiana “Midwest cohort” were assembled to create a geographically diverse cohort of career firefighters. While the cohorts were pooled, due to large differences in sample sizes, each cohort was given equal weight during the statistical analyses. Female firefighters from the Southeast (*n* = 29) and Southwest cohorts (*n* = 15) were not included in the analysis sample due to inadequate sample sizes to produce reliable estimates for these cohorts within age categories. Antihypertensive medication usage was not available for the male fighters from the Southeast cohort (*n* = 337); therefore, they were only included in analyses involving the blood pressure measurements. During data quality checks, systolic blood pressure (SBP) values were excluded if ≥182 mmHg (*n* = 4) or ≤86 mmHg (*n* = 8); diastolic blood pressure (DBP) values were excluded if >110 mmHg (*n* = 2) or <50 mmHg (*n* = 3). The sample included 5063 males and 274 females. The occupational medical evaluations reviewed in this study occurred between 2015 and 2018 and included data from questionnaires and medical evaluations. Each of the 4 clinics shared a de-identified digital database with researchers from the First Responder Health and Safety Laboratory at Skidmore College. The study protocol was reviewed and approved by the Skidmore College Institutional Review Board. Data from the publicly available 2015–2016 and 2017–2018 National Health and Nutrition Examination Surveys (NHANES) included 3004 male and 3322 female adults (20–59 years old) from the general population as a comparison group to the firefighter cohort [25]. The same cut-off values for exclusion of blood pressure values reported above were also applied to the NHANES sample.

### 2.2. Measures

Each of the 4 clinics conducted evaluations which were consistent with the National Fire Protection Association’s 1582 Standard on Comprehensive Occupational Medical Program for Fire Departments [26]. During these medical evaluations, height and weight were measured using a stadiometer and a physician’s scale, respectively, and prescription medications were identified using a questionnaire. Blood pressure was measured in a seated position by a professional examiner using auscultation technique.

Firefighters and the general population sample were grouped into age groups of 20–29 years, 30–39 years, 40–49 years, and 50–59 years. Hypertension was defined as having one or more of the following criteria: SBP ≥ 130 mmHg, DBP ≥80 mmHg, or if participants reported taking antihypertensive medications [27]. Among firefighters taking antihypertensive medications, uncontrolled hypertension was defined as SBP values ≥ 130 mmHg, or DBP ≥ 80 mmHg, and controlled hypertension was defined as SBP values < 130 mmHg, and DBP < 80 mmHg. Unmedicated hypertension was defined as having SBP ≥ 130 mmHg or DBP ≥ 80 mmHg, and not taking antihypertensive medications. The summation of the prevalence of uncontrolled hypertension, controlled hypertension, and unmedicated hypertension is equal to the prevalence of hypertension in each group. The summation of uncontrolled and controlled hypertension is equal to the prevalence of blood pressure medication usage. Body weight was categorized as normal (<25 kg·m^−2^), overweight (≥25 and <30 kg·m^−2^), and obese (≥30 kg·m^2^) based on their body mass index (BMI) [28]. Three male firefighters in the normal weight category had BMIs < 18.5 kg·m^2^.

### 2.3. Statistical Analysis

Descriptive statistics are reported as a percent or mean ± standard deviation in Table 1. Given the large differences in sample size across the clinics and the convenience nature of the fire departments who collected and provided data, all point estimates and statistical comparisons were calculated as though there were equal number of observations at each location (i.e., each geographical location contributed equally to a point estimate or the association across locations). All analyses were stratified by sex. The firefighter cohorts were combined and linear or logistic regression model outputs (e.g., predicted means and standard errors) were calculated for each age group as though there were equal number of observations in each of the geographical locations where firefighter medicals exams were obtained. The predicted means and standard errors reported were estimated with each age group modeled separately. For each outcome, trends analyses across the age groups were conducted using linear or logistic regression models with the four age groups modeled as a continuous variable. Predicted values from the combined firefighter sample were also compared to the general population using the combined NHANES data from 2015–2016 and 2017–2018. NHANES uses a complex, multistage, probability sampling design which prevents making direct statistical comparisons by combining with other datasets [29]. To compare point estimates between the combined firefighter cohort and NHANES, 83.4% confidence intervals of the predicted values in each dataset were calculated, where non-overlapping confidence intervals indicated a type 1 error of *p* < 0.05 with the assumptions that the variances were equal and estimates were independent [30]. The level of significance for all analyses was considered at *p* < 0.05 and was two-sided for all tests. All analyses were conducted using Stata 15.1 (StataCorp, College Station, TX, USA).

## 3. Results

Descriptive statistics for the combined cohort are provided in Table 1, and for each of the four firefighter cohorts in Appendix A. The mean age of male firefighters was 41.8 years and the mean age of female firefighters was 39.7 years. The percent of obese firefighters was 43% for males and 28% for females. Among the total firefighter sample, the average SBP was 124.1 ± 10.6 mmHg, average DBP was 80.9 ± 7.0 mmHg, and the prevalence of high blood pressure was 69%, with 17% taking antihypertensive medications.

Mean blood pressure values among males and female firefighters across age groups for the combined firefighter sample and the general population are presented in Figure 1. Across the age groups, SBP and DBP significantly increased with age *(p* < 0.05) among male and female firefighters. For male firefighters, mean SBP increased an average of 4.3 mmHg from the 20–29 year old group to the 50–59 year old group (120.8 ± 0.5 mmHg to 125.1 ± 0.5 mmHg) and mean DBP increased by 3.4 mmHg (77.2 ± 0.3 mmHg; 80.6 ± 0.3 mmHg). For female firefighters, mean SBP increased an average of 13.7 mmHg from the 20–29 year old group to the 50–59 year old group (113.4 ± 1.3 mmHg to 127.1 ± 2.6 mmHg) and mean DBP increased by 8.3 mmHg (72.7 ± 0.8 mmHg to 81.0 ± 1.5 mmHg). Compared to the general population, SBP was slightly higher (about 2.5 mmHg; *p* < 0.05) only among 20–29 years old male firefighters, but DBP was higher *(p* < 0.05) across all age groups for both male and female firefighters.

Percentages of male and female firefighters with controlled hypertension, uncontrolled hypertension, and unmedicated hypertension are reported in Figure 2. The percent of hypertensive firefighters increased with age (*p <* 0.001) among both male and female firefighters. Percentages of hypertensive male firefighters and female firefighters were 44.6 ± 3.2% and 10.5 ± 5.5% among 20–29 years old, and increased to 77.2 ± 1.6% and 78.8 ± 7.3% among 50–59 years old, respectively. Compared to the general population, the prevalence of hypertension was higher (*p* < 0.05) among males across all age groups (range of 10.6%–15.9% higher). For female firefighters, the prevalence of hypertension was not different from the general population except for a higher prevalence of hypertension among firefighters aged 50–59 years.

## 4. Discussion

This study is the first large-scale study (>5000 firefighters) to report blood pressure measurements and hypertension prevalence in US male and female firefighters across age groups by decade of life. The primary findings of this study are that percentage of hypertensive firefighters was 45% and 11% (males and females, respectively) among 20–29 years old, and increased to 77% and 79% among 50–59 years old; male firefighters consistently had a prevalence >40% of untreated hypertensive blood pressure values across age groups; diastolic blood pressure was significantly higher among both male and female firefighters as compared to the general population across all age groups; and male firefighters had a significantly higher prevalence of hypertension than the general population across all ages and female firefighters had higher prevalence than the general population for ages 50–59 years.

In this large group of firefighters, we found a mean SBP of 124.1 ± 10.6 mmHg and mean DBP of 80.9 ± 7.0 mmHg. Previous studies have reported similar blood pressure values in smaller cohorts of firefighters, with mean SBP values ranging from 112 to 133 mmHg and mean DBP values ranging from 76 to 85 mmHg [20,21,31,32,33,34]. Older studies examining hypertension prevalence in firefighters reported substantially lower prevalence rates of hypertension, ranging from 9% to 26% [17,20,21,24,31,32,34,35,36] as compared to 69% in the current study. A difference in how hypertension was defined (≥140/90 mmHg) previously versus current ACC/AHA guidelines (≥130/80 mmHg) [27] which were used in the current study, likely explains much of this discrepancy. These findings suggest that, while blood pressure values have remained relatively stable in the fire service over the years, the new ACC/AHA guidelines have now characterized more firefighters as being hypertensive, and thus, at a higher risk of CVD than previously appreciated. This study is one of the first to examine blood pressure in US firefighters using the newly updated ACC/AHA blood pressure guidelines. Although earlier studies reported on increased incidence of cardiac events in firefighters with high blood pressure using the previous guidelines (≥140/90 mmHg), there was convincing evidence that pre-hypertensive firefighters were also at increased risk. Kales et al. conducted a review of blood pressure in firefighters and other emergency responders and found that a majority of incidents of cardiac events occurred in responders who were pre-hypertensive using the previous guidelines (120–139/85–59 mmHg) or mildly hypertensive (140–146/88–92 mmHg) [37]. Noh et al. also found an elevated risk of sudden cardiac events in firefighters for all blood pressure values ≥ 120/80 mmHg [11]. These findings indicate that even “mildly” elevated blood pressure in firefighters is associated with increased risk of cardiac events. A firefighter suffering a cardiac event could also endanger public safety given that firefighters work in teams to perform their emergency work, and an injured or unconscious firefighter can divert attention and resources from the emergency incident.

Both male and female firefighters had a significantly higher prevalence of hypertension with increasing age. Hypertension prevalence has been shown to increase with age in the general population [27], and in 2002, using the hypertension cut-off values of that time (≥140/90 mmHg), Davis and colleagues reported a hypertension prevalence of 12.0% among young firefighters and 31.6% in older firefighters [24]. While not formally tested in our study, female firefighters had lower blood pressure values than male firefighters on average, which is consistent with previous findings [22,23]. The results of this study, however, extend previous research by showing that female firefighters’ blood pressure reached values more similar to male firefighters at older ages. Specifically, for 20–29 years old, the prevalence of hypertension among female firefighters was 11% and among male firefighters it was 45%, but for 50–59 years old, it was 79% and 77%, respectively.

Studies have also reported that a low proportion of hypertensive firefighters receive treatment and have controlled blood pressure [17,20,21,33]. Davila et al. examined the prevalence of hypertension across 13 different occupational groups and found that protective service workers (firefighters and police) had the lowest awareness of their hypertensive blood pressures (51% aware), the lowest treatment for hypertension in those who were aware (79% treated), and the lowest prevalence of controlled blood pressure in those who were treated (48% controlled) [17]. Other studies have reported the prevalence of uncontrolled hypertension (taking antihypertensive medication but blood pressure ≥140/90 mmHg) among hypertensive firefighters as 16% [20,21]. In the current study, uncontrolled hypertension (taking antihypertensive medication but blood pressure ≥130/80 mmHg) in male firefighters was 25% in 50–59 years old and 18% in female firefighters of the same age. We also found a high prevalence of unmedicated hypertension across all age groups (ranging 9–50% in females, and 43–51% in males). Unmedicated hypertension may be due to individuals not knowing they are hypertensive, not being prescribed antihypertensive medication, or not being willing to take prescribed medications. Our findings extend results from two studies with comparatively small sample sizes which reported prevalence of unmedicated hypertension in firefighters of 33% [21] and 72% [20].

SBP in firefighters did not differ from the general population for either males or females except in male firefighters 20–29 years old, and the difference of those statistically significant was less than 3 mmHg, which is of questionable clinical significance. However, DBP was significantly higher in both male and female firefighters than the general population across all age groups (ranging from 2.2 to 7.6 mmHg in males, and 3.5 to 6.3 mmHg in females). To date, no known studies have reported a difference in DBP between firefighters and the general population. In a study on the effect of working conditions on blood pressure in firefighters, Choi et al. reported that the number of 24-h shifts worked per month was related to DBP, and firefighters who performed 14 and 16 24-h shifts per month had a higher DBP (by 3.4 and 5.7 mmHg, respectively) compared to those working 8–11 shifts (reference group) [21]. The authors proposed the interruption of circadian rhythm as a potential factor attributing to this finding [21]. Sleep disruption in firefighters has also been shown to increase several CVD risk factors including hypertension [38]; however, further research is needed to specifically determine if the occupational work and exposures of firefighting impact blood pressure.

The clinical significance of higher DBP among firefighters as compared to the general population is unclear. The higher prevalence of hypertension in the current study may be attributed to the significantly higher DBP in firefighters, given that approximately 50% of firefighters in the sample who were hypertensive were defined as such based solely on elevated DBP (isolated diastolic hypertension (IDH)) classified as SBP < 130 mmHg and DBP ≥ 80 mmHg. In the general population, IDH (based on previous blood pressure standards, SBP < 140 mmHg and DBP > 90 mmHg) was not associated with increased risk of atherosclerotic cardiovascular events [39,40]. A recent analysis showed that when applying the previous or current blood pressure guidelines, IDH was not associated with subclinical or clinical atherosclerotic CVD [40]. However, in a study of 1.3 million patients, DBP was shown to be independently associated with adverse cardiovascular outcomes, regardless of the definition of hypertension, although SBP had a greater effect on outcomes [41]. IDH is also significantly associated with future systolic hypertension, with a 23-fold increased risk of developing elevated SBP in addition to elevated DBP over 10 years [42], which could suggest the importance of aggressive blood pressure surveillance and lifestyle modifications at the onset of IDH. Further research is needed to better understand IDH in the fire service.

Hypertension exacerbates atherosclerotic CVD and can lead to structural changes to the heart including LVH and cardiomegaly [6,7]. A structurally enlarged heart is associated with increased mortality risk, through various mechanisms independent of the underlying etiology [43,44]. It is also clear that adequately treating and controlling hypertension is associated with a decreased risk of cardiac events [7]. Retrospective studies in the fire service have shown that firefighters with hypertension were 3.5 times more likely to experience sudden cardiac death than their normotensive colleagues [45], and normotensive firefighters were over 4 times more likely to survive a cardiac event than hypertensive firefighters [46]. Further, a large-scale autopsy study of duty-related deaths due to cardiac causes found that firefighters with LVH and cardiomegaly were at 112 times greater risk of sudden cardiac death during fire suppression compared to station work [4].

The cause of the high prevalence of hypertension in the fire service was not specifically investigated in this study, but it may be related to occupational factors including high psychological stress [14], exposure to smoke and particulate matter [12,13], exposure to noise [15], and disrupted sleep patterns [16]. Additional research should investigate intervention strategies to lower blood pressure among firefighters and effective strategies to ensure adequate screening and proper treatment of firefighters.

### Stregnths and Limitations

This is the first study to report on blood pressure and hypertension among a large, geographically diverse group of male and female firefighters in the US. The blood pressure values were obtained by health care providers as part of a mandatory occupational health examination, so this study did not suffer from selection bias as some earlier studies may have. Despite these strengths, there are some limitations. While the sample was large and geographically diverse, it was not a random national sample. However, the general findings across the different cohorts consistently demonstrated a need for aggressive prevention measures to reduce the risk for hypertension and to aggressively manage high blood pressure throughout a firefighter’s career.

## 5. Conclusions

We reviewed occupational health records and found that a large percentage of firefighter were hypertensive based on blood pressure measurements or medication usage. The prevalence of hypertension increased with age, and was higher than the general population among males. Physicians who evaluate firefighters as part of routine occupational screening and those that serve as primary care providers should aggressively counsel firefighters on the importance of managing high blood pressure with lifestyle changes or medication.

## Figures and Tables

**Figure 1 ijerph-18-05432-f001:**
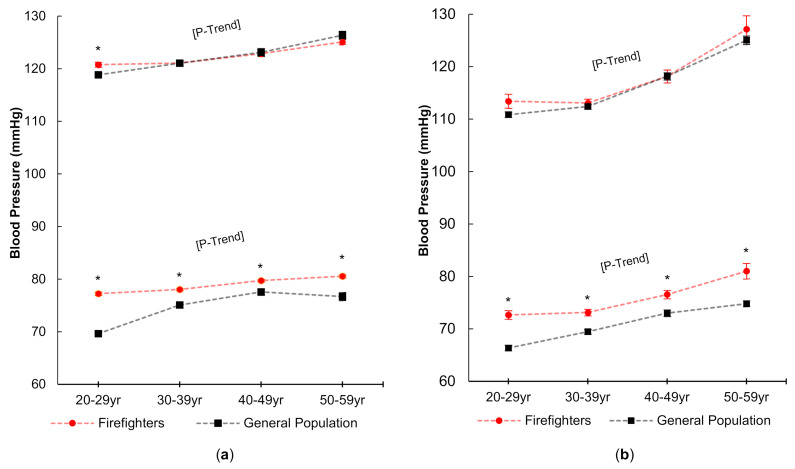
Systolic and diastolic blood pressure in males (**a**) and females (**b**) in each age group as compared to the general population (NHANES). * Indicates a significant (*p* < 0.05) difference between the combined firefighters sample and the general population. P-Trend indicates a significant increase (*p* < 0.05) across age groups in the combined firefighter sample. Standard errors of the mean are presented as error bars.

**Figure 2 ijerph-18-05432-f002:**
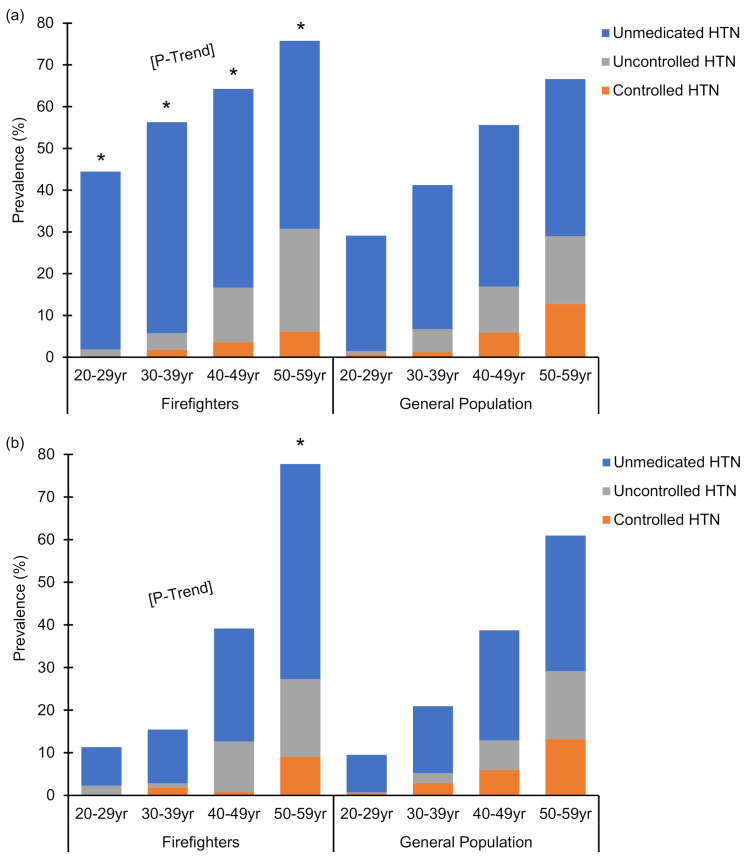
The prevalence of controlled hypertension, uncontrolled hypertension, and unmedicated hypertension within hypertensive male (**a**) and female (**b**) firefighters by age category as compared to the general population. Summation of the prevalence of controlled hypertension, uncontrolled hypertension, and unmedicated hypertension is equal to the prevalence of hypertension in each group. Summation of controlled and uncontrolled hypertension is equal to the prevalence of blood pressure medication usage. * Indicates a significant difference *(p* < 0.05) between prevalence of hypertension in the combined firefighter sample versus the general population. P-Trend indicates a significant increase *(p* < 0.05) in prevalence of hypertension across age groups in the combined firefighter sample. HTN: hypertension.

**Table 1 ijerph-18-05432-t001:** Descriptive statistics of the combined firefighter cohorts.

Characteristics	Total Sample	Males	Females
*N*	5337	5063	274
Age (y)	41.8 ± 9.2	41.9 ± 9.2	39.7 ± 8.8
White *n* (%)	4256 (86)	4025 (86)	231 (86)
African American/Black *n* (%)	373 (8)	348 (7)	25 (9)
Other *n* (%)	301 (6)	288 (6)	13 (5)
Height (m)	1.8 ± 0.1	1.8 ± 0.1	1.7 ± 0.1
Body Weight (kg)	95.2 ± 17.1	96.1 ± 16.7	77.6 ± 16.7
Body Mass Index (kg·m^−2^)	29.7 ± 4.6	29.8 ± 4.5	27.7 ± 5.3
Normal *n* (%) ^a^	691 (13)	598 (12)	93 (35)
Overweight *n* (%) ^b^	2366 (45)	2265 (45)	101 (38)
Obese *n* (%) ^c^	2221 (42)	2146 (43)	75 (28)
Systolic Blood Pressure (mmHg)	124.1 ± 10.6	124.4 ± 10.4	117.6 ± 11.7
Diastolic Blood Pressure (mmHg)	80.9 ± 7.0	81.1 ± 6.8	76.2 ± 7.9
Hypertension *n* (%) ^d^	3395 (69)	3300 (71)	95 (35)
Antihypertensive Medication *n* (%)	835 (17)	804 (17)	31 (11)
Isolated Diastolic Hypertension *n* (%) ^e^	1869 (36)	1816 (36)	53 (20)

Data presented as *n* (%) or mean ± standard deviation. ^a^ Normal BMI defined as <25 kg·m^−2^; ^b^ overweight BMI defined as 25–29.9 kg·m^−2^; ^c^ obese BMI defined as ≥30 kg·m^−2^; ^d^ hypertension defined as systolic blood pressure ≥ 130 mmHg, or diastolic blood pressure ≥ 80 mmHg, or taking antihypertensive medication; ^e^ isolated diastolic hypertension was defined as systolic blood pressure < 130 mmHg and diastolic blood pressure ≥ 80 mmHg.

## Data Availability

The data supporting the reported results are not publically available or available upon request given the agreement with the occupational health clinics that the data would only be viewed by the staff members in the First Responder Health and Safety Laboratory at Skidmore College and only used for scientific publication purposes.

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
