# Peer review of "Hypertension in the United States Fire Service"

_ijerph, 2021, doi:10.3390/ijerph18105432_

Round 1
Reviewer 1 Report
The authors assessed systemic blood pressures and prevalence of hypertension among US firefighters and the general US population. The topic is interesting. The statistical analyses are appropriate, the paper is extremely well-written and organized, and the discussion is balanced. The results presented in the graphs and tables are clear. I could not find any areas for improvement. I really enjoyed reading this manuscript.
Reviewer 2 Report
This work estimated the prevalence of hypertension in data from medical records of career male and female firefighters from 4 locations of the United States. Systolic and diastolic blood pressure mean values were compared across decades of life and between the firefighters’ sample and a sample from the general population. They found significant trends to increasing blood pressure as a function of age, no difference between the systolic blood pressure of both populations, and higher values of diastolic blood pressure in firefighters than in the general population. The firefighters’ cardiovascular health in general and hypertension are relevant issues that have been studied previously, as mentioned in the Introduction section. The current study was performed on large samples from 4 different locations and include several original results such as the comparisons by gender and by use of hypertensive medication. The manuscript is well-written. However, there are a few issues that should be addressed.
- Line 101. The body weight was defined as normal if the body mass index was below 25 kg/m2, which could include underweight subjects. Why there was no lower limit for the body mass index considered as “normal”?
- Table S1 should include the statistical comparisons between the reported groups, with significant differences indicated in the table.
- Line 211. The discussion about the influence of hypertension on the increased risk of developing cardiovascular disease is interesting, including the importance of assessing hypertension with lower cut-off values as the current ACC/AHA guidelines recommend, to identify and treat people with hypertension earlier. However, the last phrase seems to be overreaching about the consequence of the risk of cardiac events in firefighters: “…which could also endanger public safety given the work they perform”. I recommend adding appropriate references to support such a statement or otherwise to delete it.
- Conclusions section: Although a brief statement about the implications of the findings of the present study are valid (and desirable), this section should address mainly the actual findings of the present work, instead of general recommendations. Please update this section accordingly.
